# NetPRS: SNP interaction aware network-based polygenic risk score for Alzheimer's disease

Sunghong Park
*Department of Physiology*
*Ajou University School of Medicine*
Suwon, Republic of Korea
pshong513@ajou.ac.kr

Dong-gi Lee
*DBEI, Perelman School of Medicine*
*University of Pennsylvania*
Philadelphia, United States
donggi.lee@pennmedicine.upenn.edu

Juhyeon Kim
*Department of Industrial Engineering*
*Ajou University*
Suwon, Republic of Korea
juhyeon@ajou.ac.kr

Doyoon Kim
*Department of Physiology*
*Ajou University School of Medicine*
Suwon, Republic of Korea
kdy1249@ajou.ac.kr

Chang Hyung Hong
*Department of Psychiatry*
*Ajou University School of Medicine*
Suwon, Republic of Korea
antiaging@ajou.ac.kr

Sang Joon Son
*Department of Psychiatry*
*Ajou University School of Medicine*
Suwon, Republic of Korea
sjsonpsy@ajou.ac.kr

Hyun Woong Roh
*Department of Psychiatry*
*Ajou University School of Medicine*
Suwon, Republic of Korea
hansin8607@naver.com

Kanghee Park
*Tech. Intelligence Research Team*
*KISTI*
Seoul, Republic of Korea
can17@kisti.re.kr

Dokyoon Kim
*DBEI, Perelman School of Medicine*
*University of Pennsylvania*
Philadelphia, United States
dokyoon.kim@pennmedicine.upenn.edu

Hyunjung Shin
*Department of Industrial Engineering*
*Ajou University*
Suwon, Republic of Korea
shin@ajou.ac.kr

Hyun Goo Woo
*Department of Physiology*
*Ajou University School of Medicine*
Suwon, Republic of Korea
hg@ajou.ac.kr

*Abstract*—**Alzheimer's disease (AD) is underscored by its polygenic nature, attributable to variants across multiple genetic loci. This has led to the development of the polygenic risk score (PRS) model, which estimates individual risk by aggregating risk alleles weighted from their effect sizes. While early models were limited to utilizing only independent effects of single nucleotide polymorphisms (SNPs), recent models have been advanced to consider epistatic interactions between SNPs. However, SNPs interact through various channels, and typically, they are associated with each other through SNP-gene relations and gene-gene interactions. Moreover, SNPs interact synergetically, exhibiting diverse joint effects of genetic variations. Given these properties of SNP interactions, the PRS models need improvement to account for the interactive effects between SNPs in a polygenic manner, especially for genetically complex diseases such as AD. In this study, we propose a two-stage approach for AD risk assessment, called network-based PRS (NetPRS). First, the phenotypic and genomic interactions are quantified and integrated into networks. Second, the independent effects of SNPs are propagated on the integrated SNP networks using graph-based machine learning model. Through this procedure, NetPRS extracts the globally interactive effects between SNPs and integrates these effects to predict the risk of AD. The proposed method was applied to two cohort datasets: the Alzheimer's Disease Neuroimaging Initiative dataset with 1,175 participants, and a South Korean dataset with 724 participants. Experimental results showed that the integrated effects of NetPRS more clearly distinguished between AD and control groups, outperforming the six existing methods by 16.4% on average.**

*Keywords*—*Polygenic risk score, Alzheimer's disease, SNP interaction network, semi-supervised learning (SSL)*

This study was conducted with biospecimens and data from the consortium of the Biobank Innovations for Chronic cerebrovascular disease With ALZheimer's disease Study (BICWALZS), which was funded by the Korea Disease Control and Prevention Agency for the Korea Biobank Project (#6637-303). This work was supported by the Basic Science Research Program through the National Research Foundation of Korea (NRF) funded by the Ministry of Education (MOE), Republic of Korea (NRF-2022R1A6A3A01086784), the BK21 FOUR program of the NRF funded by the MOE (NRF5199991014091), and Ajou University Research Fund. This study was also supported by the NRF grants funded by the Ministry of Science and ICT (MSIT), Republic of Korea (NRF-2019R1A5A2026045, NRF-2021R1A2C2003474, and NRF-RS-2022-001653), a grant funded by the MSIT (KISTI Project No.K24L3M1C2), the Institute of Information & communications Technology Planning & Evaluation (IITP) grants funded by the MSIT (IITP-2024-No.RS-2023-00255968 for the Artificial Intelligence Convergence Innovation Human Resources Development and No. 2022-0-00653), the Korea Health Technology R&D Project through the Korea Health Industry Development Institute (KHIDI) funded by Ministry of Health and Welfare (MOHW), Republic of Korea (HR21C1003), a grant of 'Korea Government Grant Program for Education and Research in Medical AI' through the KHIDI funded by the Korea government (MOE and MOHW) and a grant funded by the National Institutes of Health, USA (R01 AG071470). (Sunghong Park, Dong-gi Lee, and Juhyeon Kim contributed equally to this work.) (Corresponding authors: Hyunjung Shin; Hyun Goo Woo.)

## I. INTRODUCTION

Alzheimer's disease (AD) is one of the most common neurodegenerative disorders, particularly affecting elderly people over 65 years of age with about 10% prevalence [1]. With the rapid aging of the global population, AD is becoming more prevalent, and the number of patients is expected to increase from 50 million in 2015 to 130 million in 2050 [2]. As AD has recently emerged as a serious problem, numerous studies have been conducted to overcome AD, focusing on achieving accurate diagnosis and early prediction. A representative field is the study based on genetic data. Genetic factors play a key role in AD pathogenesis, highlighting the importance of using genetic data to predict AD risk [3]. Recent genome-wide association studies (GWAS) have revealed that multiple genetic loci contribute to AD pathogenesis [4, 5], expanding genetic risk assessment for AD to polygenic

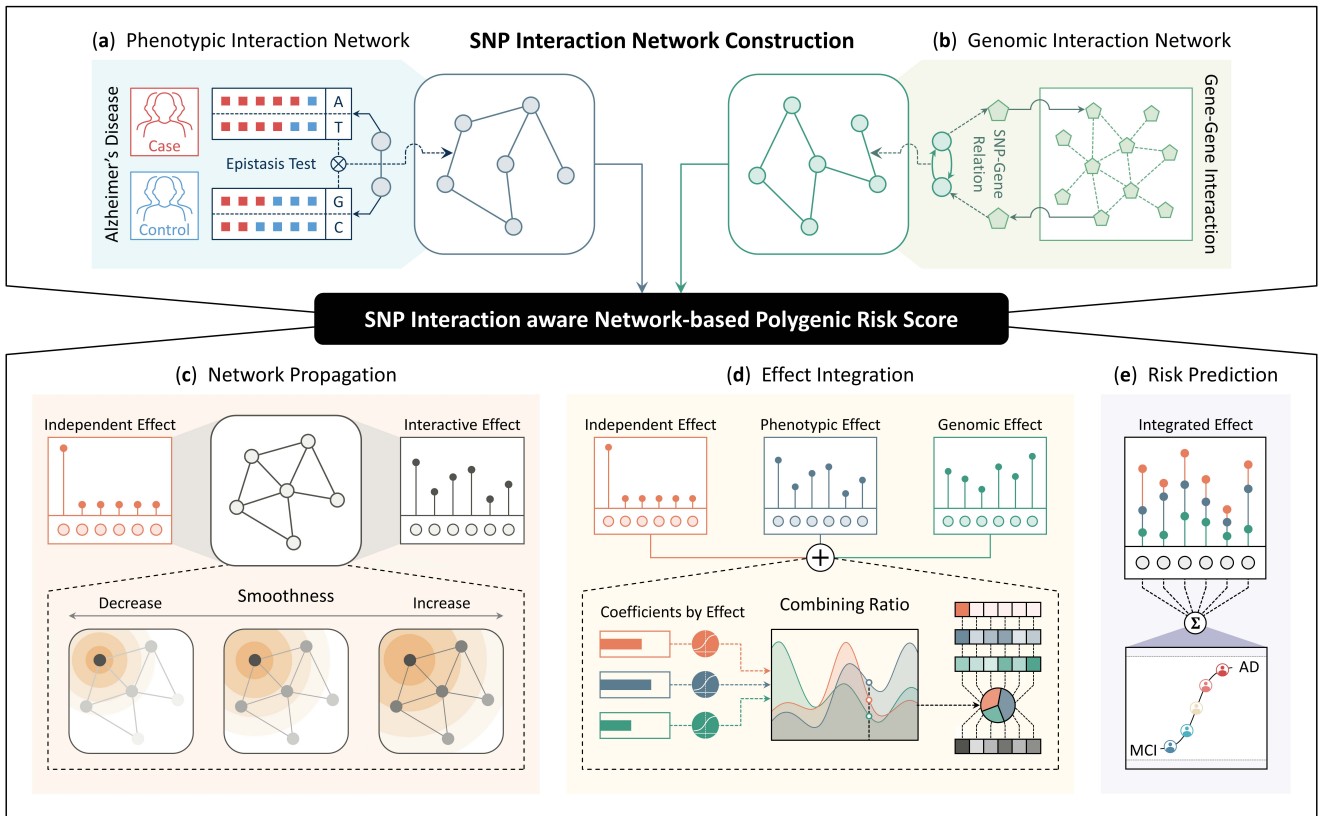

Fig. 1. Schematic description of the proposed method. The proposed method consists of two stages: SNP interaction network construction and network-based polygenic risk score (NetPRS). At first, the proposed method quantifies the phenotypic and genomic interactions between SNPs and represent them as networks. (a) Phenotypic SNP interactions are measured by the epistasis test. (b) Genomic SNP interactions are measured by applying SNP-gene relations to gene-gene interaction network. Next, NetPRS assesses individual AD risk by utilizing the independent effect and the constructed SNP interaction networks. (c) Network propagation extracts the phenotypic and genomic interactive effects between SNPs by diffusing the independent effect to each network. Smoothness is a trainable parameter that controls the intensity of diffusion. As its value decreases, interactions be-tween close SNPs are highlighted, and as its value increases, even interactions between distant SNPs are reflected. (d) Effect integration linearly combines the independent effect and the extracted two interactive effects into the integrated effect. Combining coefficients are set as trainable parameters indicating the ratio of each effect reflected in the integrated effect. (e) Risk prediction calculates individual AD risk by applying the integrated effect to the parameter representing the effect size of SNPs.

methods that consider other genetic factors in addition to APOE, which is known to be the most important AD-related gene [6]. Therefore, polygenic score models that integrate the effects of multiple genetic variants identified through GWAS have emerged as a key method for effectively predicting AD risk [7].

The polygenic risk score (PRS) model is the most commonly used method, which is constructed by calculating the sum of risk allele dosages weighted by their GWAS-derived effect sizes [8]. This approach was applied to individual assessment of AD risk [9] and investigated with respect to the optimal strategy related to genetic data processing that can achieve the highest accuracy [10]. Although the PRS-based assessment is an appropriate method to reflect the polygenic nature of AD, the challenge remains that the PRS model is limited to the independent effects of SNPs without considering the interactive effects between SNPs [11]. The interactions between SNPs have a critical impact on disease onset and progression through various pathways of genetic variation. In AD pathogenesis, SNP interactions can promote or inhibit pathological processes such as amyloid beta protein accumulation in the brain or abnormal phosphorylation of tau protein [12, 13]. The fact that the SNP interactions play a key role in understanding the

complex genetic basis of Alzheimer's disease and developing personalized prediction models suggests that the PRS model needs to be further sophisticated by reflecting the interactive effects between SNPs.

The epistatic effect is the most representative SNP interaction, which indicates the impact on the traits of individuals caused by the interactive effects occurring at different genetic loci. As the interactions between SNPs by epistasis significantly contribute to determining phenotype [14], the analysis of epistatic effects is important for understanding the mechanisms of disease pathogenesis [15]. The advantage of epistatic effect can be more beneficial for complex diseases such as AD. Several AD-specific key genes were identified by utilizing the epistasis-based SNP interactions [16], and the accuracy of the PRS model for AD was improved by combining the independent effect with the interactive effect between SNPs by the epistasis [17]. As a result, the PRS model achieves a more in-depth understanding of AD pathogenesis and a more accurate AD risk assessment by considering SNP interaction effects by epistasis.

In addition to the epistatic effect, SNPs also interact with the genome, and such interactions can act as a stronger risk factor for disease by working synergistically [18]. The SNP interactions based on the genome, such as enhancer-promoter

interactions, are significantly related to disease risk by affecting the regulation of gene expression. [19]. Furthermore, the genomic SNP interactions occur not only within the same gene but also across different genes. Recent studies have shown that such interactions are considered important risk factors for various diseases [20, 21]. Especially, AD is a complex disease involving the interactions between various molecular pathways, regarding the effects of genomic interactions more important [22]. Consequently, the effectiveness of genomic interactions on disease onset and the complexity of AD pathogenesis highlights the need for the PRS model to consider the genomic interactions for the AD risk assessment.

For the more comprehensive and sophisticated assessment of AD risk, the best strategy would be for the PRS model to consider both phenotypic and genomic SNP interactions. However, there are several challenges in implementing this strategy as follows. *(i) Quantification of genomic SNP interactions*: while there is abundant information on SNP-gene relations, there is a lack of information on the interactive effects that SNPs exert through their related genes. *(ii) Computation of exponentially increasing SNP interactions*: when the PRS model considers SNP interactive effects as input data, the number of variables increases exponentially, resulting in enormous computational complexity. Moreover, SNP interactions extend to higher orders. Although utilizing the synergistic effects of more SNPs helps in the more precise AD risk assessment, calculating the effect size becomes more exponentially complex. *(iii) Integration of multiple SNP effects*: by including not only independent effects of SNPs but also phenotypic and genomic interactive effects in the PRS model, AD risk can be calculated from a comprehensive perspective. At this time, the impact of each effect on AD may not be the same. Therefore, when utilizing various effects in an integrated manner, it is necessary to assign different weights rather than simple merging, which suggests that the PRS model should derive optimized weights for each effect.

To overcome the aforementioned challenges, we propose a novel method, *SNP interaction aware network-based polygenic risk score*, for AD risk assessment. As shown in Fig. 1, the proposed method consists of two stages: *SNP interaction network construction* and *network-based polygenic risk score (NetPRS)*. In the first stage, the proposed method quantifies not only the phenotypic interactions by the epistasis test as Fig. 1(a) but also the genomic interactions between SNPs as Fig. 1(b). By applying SNP-gene relations to gene-gene interactions (GGIs), we consider the indirect genomic interactions between SNPs that can occur through GGIs. This approach is not limited to simply counting the number of shared genes, but also quantifies the genomic SNP interactions that comprehensively reflect the properties of GGIs. The quantified SNP interactions are represented as individual networks. In the second step, NetPRS derives interactive effects between SNPs based on the constructed networks and integrates all effects to predict AD risk. To implement this process, NetPRS consists of *network propagation*, *effect combination*, and *risk prediction*, which execute sequentially. The first function, network propagation, extracts the phenotypic and genomic interactive effects based on the constructed two networks. Fig. 1(c) shows that each interactive effect is obtained by diffusing the independent effect throughout each network, so that NetPRS involves the global interactions between SNPs. Additionally, the intensity of diffusion is controlled by the smoothness parameter, which

is trainable. As the smoothness decreases, interactions between close SNPs are highlighted, and as the smoothness increases, even interactions between distant SNPs are reflected. Therefore, NetPRS can scale SNP interactions to infinite-order that are differentially reflected by order. Next, the second function, effect integration, combines three effects into the integrated effect, as shown in Fig. 1(d). The integrated effect is represented by the linear combination of all effects. Combining coefficient for each effect indicates the ratio of each effect reflected in the integrated effect. The coefficients are optimized through model training, and three effects differently contribute to the best AD risk assessment. In Fig. 1(e), the third function, risk prediction, receives the integrated effect as input data and calculates individual AD risk, as depicted. Finally, the proposed method is summarized as the individual AD risk assessment by utilizing the independent effect and the phenotypic and genomic interactive effects applying global interactions between SNPs.

## II. SNP INTERACTION NETWORK CONSTRUCTION

### A. Phenotypic Interaction Network

To quantify the phenotypic SNP interactions, we perform the epistasis test using a GWAS tool, PLINK [23]. The epistasis test involves applying a logistic regression model to categorize case and control groups according to allele dosage from two SNPs, denoted as $S^j$ and $S^k$, in the following manner:

$$\log \frac{Pr(Y=1|S^j,S^k)}{Pr(Y=0|S^j,S^k)} = \beta_0 + \beta_1 S^j + \beta_2 S^k + \beta_3 S^j S^k + e \quad (1)$$

where $Y$ represents the control group if it takes the value of 0 and the case group if it takes 1. The interaction between $S^j$ and $S^k$ is calculated by the odds ratio, denoted as $h_P^{jk}$, which is derived from the coefficients of the model in (1), and specifically, it focuses on the coefficient $\beta_3$ for the interaction. If the value of $h_P^{jk}$ is 1, it indicates that there is no interaction between $S^j$ and $S^k$. The further the value deviates from 1, the stronger the interaction between $S^j$ and $S^k$. In alignment with this concept, $h_P^{jk}$ is transformed by the function $T_P$, which converts its value into a measure of similarity between SNPs, as below.

$$T_P(x) = -e^{-|\ln x|} + 1$$

Then, the phenotypic interaction between $S^j$ and $S^k$ is calculated as $W_P^{jk} = T_P(h_P^{jk})$, and this procedure is applied to every pair of SNPs. At last, after thresholding the interactions according to their significance, the phenotypic interaction network $W_P \in \mathbb{R}^{s \times s}$ is constructed, where $s$ is the number of SNPs.

### B. Genomic Interaction Network

The genomic interaction network, denoted as $W_G \in \mathbb{R}^{s \times s}$, is constructed by combining the SNP-gene relation ($g \in \mathbb{R}^{m \times s}$) with the GGI network ($W \in \mathbb{R}^{m \times m}$), where $m$ is the number of genes. Denoting $h_G^{jk}$ as the genomic interaction from $S^j$ to $S^k$, $h_G^{jk}$ indicates the effect for $S^j$-related genes ($g^j \in \mathbb{R}^{m \times 1}$) on $S^k$-related genes ($g^k \in \mathbb{R}^{m \times 1}$) through GGI. By representing $f^j \in \mathbb{R}^{m \times 1}$ for the effect of $g^j$ on GGI, the objective function for $f^j$ is defined by applying graph-based semi-supervised learning (GSSL) [24] as follows:

$$\min_f (f^j - g^j)^T (f^j - g^j) + \mu f^{j^T} L f^j \quad (2)$$

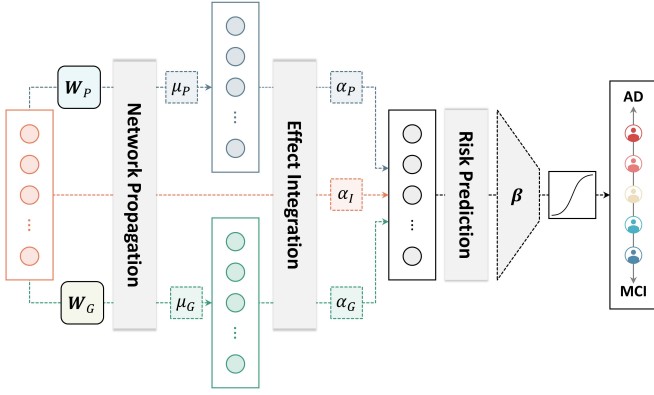

Fig. 2. Schematic structure of NetPRS. Network propagation extracts the interactive effect between SNPs by diffusing the independent effect to the SNP interaction network. The intensity of diffusion is controlled by the smoothness parameter, which is denoted as $\mu_P$ and $\mu_G$ for the phenotypic and genetic interaction network, respectively. Effect integration linearly combines the independent effect and the two interactive effects into the integrated effect. Combining coefficients, denoted as $\alpha_I$, $\alpha_P$, and $\alpha_G$ for each effect, are parameters of effect integration and are converted to probability form to determine the ratio of each effect in the integrated effect. Risk prediction calculates individual AD risk by applying the integrated effect to the SNP effect size parameter $\beta$. The structure of NetPRS is an end-to-end neural network.

where $L$ is the graph Laplacian matrix defined as $L = D - W$, $D = \text{diag}(D^a)$ and $D^a = \sum_b W^{ab}$, and $\mu$ is a hyperparameter. The solution of (2) is obtained in the closed form as

$$f^j = (I + \mu L)^{-1} g^j. \tag{3}$$

Therefrom, $h_G^{jk}$ is derived by $h_G^{jk} = f^{j^T} g^k$, and then, its value is transformed by the function $T_G$ with a scaling parameter $\sigma$ as below.

$$T_G(x) = e^{(\ln x/\sigma^2)}$$

Consequently, the genomic interaction from $S^j$ to $S^k$ is calculated as $W_G^{jk} = T_G(h_G^{jk})$, and this procedure is applied to every pair of SNPs. Finally, the genomic interaction network $W_G \in \mathbb{R}^{s \times s}$ is constructed by thresholding the interactions according to their values.

## III. NETWORK-BASED POLYGENIC RISK SCORE

NetPRS consists of three functions: *network propagation*, *effect integration*, and *risk prediction*. First, the network propagation extracts the SNP interactive effects by applying the independent effect to the phenotypic and genomic interaction networks. Second, the effect integration linearly combines the independent effect and the extracted phenotypic and genomic interactive effects into the integrated effect. Finally, the risk prediction calculates individual AD risk by applying parameters for SNP effect size to the integrated effect. As shown in Fig. 2, NetPRS is an end-to-end model designed with a neural network structure.

### A. Network Propagation

Let the data matrix of the SNP independent effect be denoted as $X \in \mathbb{R}^{s \times n}$, indicating the genotype of subjects, where $n$ is the number of subjects. To extract the interactive effects based on the phenotypic and genomic interaction networks, which are denoted as $F_P$ and $F_G$, respectively, GSSL is applied to propagate $X$ through each network.

Similar to (2), the objective function for $F_*$ is defined as, follows:

$$\min_{F_*}(F_* - X)^T(F_* - X) + \mu_* F_*^T L_* F_*$$

where $L_*$ is the graph Laplacian matrix from $W_*$, and $\mu_*$ is the trainable smoothness parameter that trades off the loss (the first term) and the smoothness (the second term). In the same manner in (3), the phenotypic and genomic interactive effects between SNPs are obtained as below.

$$F_* = (I + \mu_* L_*)^{-1} X \in \mathbb{R}^{s \times n} \tag{4}$$

### B. Effect Integration

The SNP independent effect and the two extracted interactive effects are represented as the integrated effect through a linear combination. The combining coefficients for $X$, $F_P$, and $F_G$ are denoted as $\alpha_I$, $\alpha_P$, and $\alpha_G$, respectively. These coefficients are transformed into the combining ratio, represented as $\theta_I$, $\theta_P$, and $\theta_G$, using the softmax function.

$$\theta_* = e^{\alpha_*}/\sum_k e^{\alpha_k}$$

This transformation aims to prevent the problem of combining coefficients turning from negative during the model training. Then, $Z$ is derived by linearly combining each effect with its ratio by representing $F_I$ as same as $X$.

$$Z = \theta_I F_I + \theta_P F_P + \theta_G F_G \in \mathbb{R}^{s \times n} \tag{5}$$

### C. Risk Prediction

Finally, individual AD risk is calculated by applying the integrated effect to the coefficient vector, which is denoted as $\beta \in \mathbb{R}^{s \times 1}$, indicating the SNP effect size. Since PRS provides disease risk in the form of probability, NetPRS employs a logistic classifier to assess individual AD risk.

$$P = 1/(1 + e^{-\beta^T Z}) \in \mathbb{R}^{1 \times n}$$

### D. Model Optimization

As shown in Fig. 2, NetPRS contains a total of six parameters: $\{\mu_P, \mu_G\}$ for network propagation, $\{\alpha_I, \alpha_P, \alpha_G\}$ for effect combination, and $\beta$ for risk prediction. By denoting the actual diagnosis as $Y \in \mathbb{R}^{1 \times n}$ indicating whether AD or not as 1 or 0, respectively, the parameters are trained to minimize the cross-entropy loss $\mathcal{L}$ between $P$ and $Y$ as below.

$$\mathcal{L} = -\frac{1}{n}(Y^T \log P + (1 - Y)^T \log(1 - P))$$

Then, the objective function of NetPRS is defined as follows:

$$\underset{\mu_*, \alpha_*, \beta}{\text{argmin}} \, \mathcal{L} + \delta \mathcal{R} \tag{6}$$

where $\mathcal{R}$ is the L2 regularizer with the coefficient $\delta$. The objective function is optimized by the gradient descent method.

**Minimization over $\beta$:** the gradient with respect to $\beta$ is obtained by differentiating the objective function in (6) by $\beta$.

$$\nabla \beta = \frac{1}{n} Z(P - Y)^T + 2\delta\beta$$

**Minimization over $\alpha_*$:** to find the gradient of $\alpha_*$, the derivative of $\theta_*$ with respect to $\alpha_*$ is firstly obtained according to the derivative of the softmax function as below:

$$\frac{\partial \theta_i}{\partial \alpha_i} = \theta_i(1 - \theta_i), \ \frac{\partial \theta_{j,k}}{\partial \alpha_i} = -\theta_i \theta_{j,k} \quad (7)$$

where $\{i, j, k\}$ is equivalent to the permutation of $\{I, P, G\}$. Then, by (7), the derivative of $\boldsymbol{Z}$ with respect to $\alpha_i$ is represented as follows.

$$\frac{\partial \boldsymbol{Z}}{\partial \alpha_i} = \theta_i \left( (1 - \theta_i)\boldsymbol{F}_i - \theta_j \boldsymbol{F}_j - \theta_k \boldsymbol{F}_k \right) = \theta_i (\boldsymbol{F}_i - \boldsymbol{Z}) \quad (8)$$

Therefrom, the gradient of $\alpha_*$ is derived by combining $\partial \mathcal{L}/\partial \boldsymbol{Z}$ with (8).

$$\nabla \alpha_* = \frac{\theta_*}{n} \text{Tr}\left( (\boldsymbol{F}_* - \boldsymbol{Z})^{\text{T}} \boldsymbol{\beta}(\boldsymbol{P} - \boldsymbol{Y}) \right) + 2\delta \alpha_*$$

**Minimization over $\mu_*$:** to find the gradient with respect to $\mu_*$, the derivative of $\boldsymbol{F}_*$ with respect to $\mu_*$ is firstly obtained as below:

$$\frac{\partial \boldsymbol{F}_*}{\partial \mu_*} = \frac{\partial (\boldsymbol{I} + \mu_* \boldsymbol{L}_*)^{-1} \boldsymbol{X}}{\partial \mu_*}. \quad (9)$$

By replacing $(\boldsymbol{I} + \mu_* \boldsymbol{L}_*)$ with $\boldsymbol{Q}_*$ for the brief description, (9) is derived according to the derivative of an inverse matrix [25] as follows:

$$\frac{\partial \boldsymbol{Q}_*^{-1}}{\partial \mu_*} \boldsymbol{X} = -\boldsymbol{Q}_*^{-1} \boldsymbol{L}_* \boldsymbol{Q}_*^{-1} \boldsymbol{X}. \quad (10)$$

Therefrom, the gradient of $\mu_*$ is obtained by combining $\partial \mathcal{L}/\partial \boldsymbol{F}_*$ with (10).

$$\nabla \mu_* = -\frac{\theta_*}{n} \text{Tr}\left( \left( \boldsymbol{Q}_*^{-1} \boldsymbol{L}_* \boldsymbol{Q}_*^{-1} \boldsymbol{X} \right)^{\text{T}} \left( \boldsymbol{\beta}(\boldsymbol{P} - \boldsymbol{Y}) \right) \right) + 2\delta \mu_*$$

## IV. EXPERIMENTS

### A. Data Description

As delineated in Table I, three types of data were utilized in this study. We firstly collected two genotype datasets from the Alzheimer's Disease Neuroimaging Initiative (ADNI) and the Biobank Innovations for chronic Cerebrovascular disease With Alzheimer's disease Study (BICWALZS) at Ajou University Hospital (Suwon, Republic of Korea) [26]. A total of 1,175 participants were included in the ADNI dataset, with 669,629 SNPs genotyped using the Illumina Omni 2.5M BeadChip, while the BICWALZS dataset comprised 724 participants, with 827,783 SNPs genotyped using the KoreanChip, which was designed for genomic research within the Korean population [27]. We also collected the data of approximately 334 million SNP-gene relations and approximately 6 million GGIs from the dbSNP [28] and STRING [29], respectively. In addition, we divided the participants of the two datasets into the discovery and validation cohorts for the independent validation of the proposed method. In the ADNI dataset, the discovery and validation cohorts consisted of 765 participants from ADNI 2 and 410 participants from ADNI 1/GO, respectively. In the BICWALZS dataset, 547 participants recruited between 2016 and 2019 were included in the discovery cohort comprised, and the validation cohort contained 177 participants recruited between 2020 and 2021. Table II provides the demographic characteristics of study participants by cohorts.

### B. Results for SNP Interaction Network Construction

We performed quality control (QC) according to the following criteria [30]: genotyping rate >0.99, Hardy-Weinberg Equilibrium $P$ >1E−6, and minor allele frequency >5%. In addition, Linkage Disequilibrium pruning was performed with thresholds (window size: 50, step size: 5, and

TABLE I. DATA DESCRIPTION FOR THIS STUDY

| Data | Source | Description |
|---|---|---|
| Participant genotype | ADNI (1/GO/2) | 669,629 SNPs of 1,175 participants |
| | BICWALZS | 827,783 SNPs of 724 participants |
| SNP-Gene relation | dbSNP | 333,845,887 relations between 817,918 genes and 312,846,021 SNPs |
| Gene-Gene interaction | STRING | 5,969,249 interactions between 19,566 genes |

TABLE II. DEMOGRAPHIC CHARACTERISTICS OF STUDY PARTICIPANTS

| Dataset | Characteristics | Total participants | Discovery cohort | Validation cohort |
|---|---|---|---|---|
| ADNI | Participants, No. | 1,175 | 765 | 410 |
| | AD-positive, No. (%) | 406 (34.6) | 263 (34.4) | 143 (34.9) |
| | Age, median (IQR) | 77 (72-83) | 76 (71-82) | 78 (73-84) |
| | Female, No. (%) | 537 (45.7) | 365 (47.7) | 172 (42.0) |
| BICWALZS | Participants, No. | 724 | 547 | 177 |
| | AD-positive, No. (%) | 167 (23.1) | 129 (23.6) | 38 (21.5) |
| | Age, median (IQR) | 73 (67-78) | 73 (68-78) | 72 (66-79) |
| | Female, No. (%) | 481 (66.4) | 364 (66.5) | 117 (66.1) |

$R^2$ threshold: 0.3). There were 583,526 and 399,129 SNPs that passed QC for the ADNI and BICWALZS datasets, respectively. Subsequently, we evaluated the statistical significance of QC-passed SNPs through GWAS and divided SNPs into three levels according to their $P$−values. In the ADNI dataset, 974 SNPs were screened at level 1, 130 SNPs at level 2, and 15 SNPs at level 3; in the BICWALZS dataset, 420, 63, and 13 SNPs were screened at levels 1, 2, and 3, respectively.

Then, we constructed the phenotypic and genomic interaction networks for each level of SNPs, where the phenotypic interactions between SNPs were measured by the epistasis test and selected by $P$-value of 0.05, while the genomic interactions between SNPs were measured by (3) with the smoothness of 1 and selected by the elbow point of $5.35 \times 10^{-5}$. Table III presents the summarized results for the constructed SNP interaction networks. The average density across all networks was 4.2%, with the phenotypic interaction network having an average density of 4.4%, which is 10.5% higher than that of the genomic interaction network of 3.9%. Comparing two types of networks, there were common edges in level 1 and level 2, and their correlation coefficients for edge weights indicated an average of 0.0962 for level 1 and 0.6755 for level 2, but the distributions of common edges were revealed to be statistically different by the Kolmogorov-Smirnov test, suggesting that the two types of networks represent distinct patterns of SNP interactions.

### C. Performance Evaluation

The proposed method was applied to all levels of SNP datasets. $\mu_*$ and $\alpha_*$ were initially set to 1 and 0, respectively. NetPRS was trained with a learning rate of 0.01 using ADAM optimizer [31]. By measuring the area under the receiving operating characteristic curve (AUC), the performance of NetPRS was compared with the six existing methods: wPRS

TABLE III. RESULTS FOR SNP INTERACTION NETWORK CONSTRUCTION

| Dataset | Level of SNPs (*P*-value threshold) | No.Nodes | Phenotypic Interaction Network | | | Genomic Interaction Network | | | Common SNP-SNP Interactions | | |
|---|---|---|---|---|---|---|---|---|---|---|---|
| | | | No.Edges | Avg.Edges | Density | No.Edges | Avg.Edges | Density | No.Edges | Corr.Coef. | *P*-value |
| ADNI | Level 1 ($-\log_{10}P > 3$) | 974 | 23,177 | 0.6698 | 4.8912% | 17,978 | 0.3180 | 3.7940% | 1,642 | 0.0841 | 2.4E–122 |
| | Level 2 ($-\log_{10}P > 4$) | 130 | 429 | 0.7026 | 5.1163% | 448 | 0.7362 | 5.3429% | 38 | 0.5774 | 2.5E–6 |
| | Level 3 ($-\log_{10}P > 5$) | 15 | 7 | 0.6921 | 6.6667% | 6 | 0.9787 | 5.7143% | - | - | - |
| BICWALZS | Level 1 ($-\log_{10}P > 4$) | 420 | 3,253 | 0.6505 | 3.6970% | 2,770 | 0.2653 | 3.1481% | 79 | 0.1083 | 4.2E–71 |
| | Level 2 ($-\log_{10}P > 5$) | 63 | 62 | 0.7181 | 3.1746% | 60 | 0.9182 | 3.0722% | 3 | 0.7735 | 7.7E–3 |
| | Level 3 ($-\log_{10}P > 6$) | 13 | 2 | 0.7057 | 2.5641% | 2 | 0.9948 | 2.5641% | - | - | - |

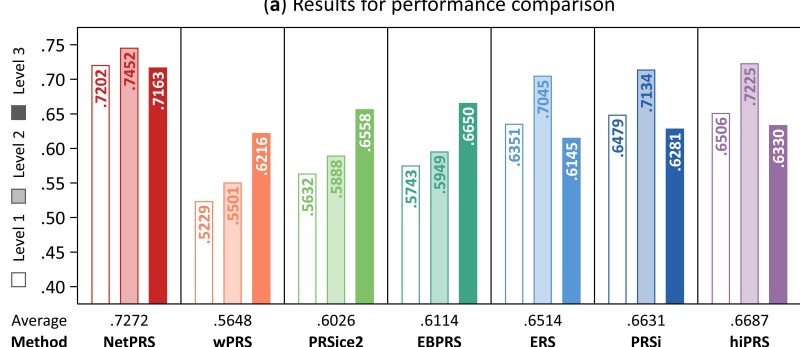

(a) Results for performance comparison

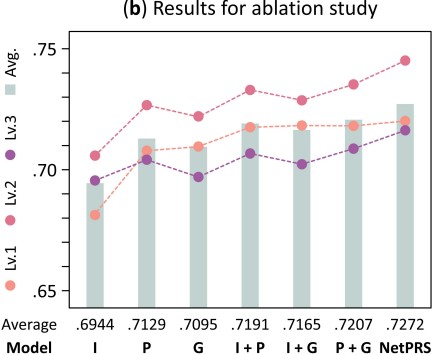

(b) Results for ablation study

Fig. 3. Results for performance evaluation on NetPRS. (a) shows the performance comparison results for the proposed and exsiting methods, illustrating the average of AUCs for AD risk prediction by applying each method on the ADNI and BICWALZS cohorts. (b) indicates the results for ablation study on NetPRS, representing the contribution of each effect to performance by training the combinatorically configured effects.

[8], PRSice2 [32], EBPRS [33], ERS [17], PRSi [34], and hiPRS [35], where the initial three methods employed the independent effects of SNPs, whereas the subsequent three methods utilized the epistatic SNP interaction, and the results are presented in Fig. 3 and the Appendix.

Fig. 3(a) compares the average of AUCs for AD prediction by applying each method on the ADNI and BICWALZS cohorts. The average of AUCs in all levels for NetPRS was 0.7272, which was an average of 16.4% higher than the comparison methods. NetPRS demonstrated the best performance at level 2, with levels 1 and 3 exhibiting comparatively lower performance. This pattern was also observed in methods based on the epistatic SNP interactions (ERS, PRSi, and hiPRS), while methods based on SNP independent effects (wPRS, PRSice2, and EBPRS) typically yielded the best performance at level 3, followed by levels 1 and 2. As a result, the independent effect-based PRS models demonstrated superior performance as more significant SNPs for AD were screened, while the interactive effect-based PRS models necessitated a data-specific threshold of significance level for the best outcomes; experimentally, the optimal performance was achieved using 0.02% of the number of QC-passed SNPs as the thresholding criteria.

In addition, we conducted an ablation study to further investigate the contribution of the three effects included in the integrated effect on the performance of NetPRS. We designed six different models that train the combinatorically configured datasets for three effects: three models for the single effect ($\Phi_I$, $\Phi_P$, and $\Phi_G$) and other three models for the dual effects ($\Phi_{I+P}$, $\Phi_{I+G}$, and $\Phi_{P+G}$). The experimental settings for those models were same as NetPRS ($\Phi_{I+P+G}$), and Fig. 3(b) shows the results. The average of AUCs in all levels for NetPRS was the

highest, followed by $\Phi_{P+G}$ and $\Phi_{I+P}$ with almost similar performance. The performance of $\Phi_I$ was the lowest, but when combined with interactive effects ($\Phi_{I+P}$ and $\Phi_{I+G}$), it improved by 3.4% on average. Among the interactive effects, $\Phi_P$ outperformed $\Phi_G$, and when they were combined, $\Phi_{P+G}$ derived 1.3% better prediction results than each. Overall, the performance was improved as the three effects were combined together. These results confirm that the interactions between SNPs are effective in predicting AD risk, and that it is more advantageous to consider not only the commonly used phenotypic interactive effects but also the newly proposed genomic interactive effect.

*D. Model Interpretation*

To interpret the proposed method, we conducted the statistical and explanatory analyses on NetPRS in level 2, which was the most accurate. First, we compared the statistical significance of the independent effect ($X$) and the combined effect ($Z$) for the AD-positive and AD-negative groups. Fig. 4(a) represents the comparison results on *P*-values of $X$ and $Z$, denoted as $P_X$ and $P_Z$, respectively. A point in the scatter plot above the diagonal line means that the SNP effect values on the vertical axis is more significant. As the plot illustrates, the majority of the dots are situated above the diagonal line, indicating that $Z$ is more effective than $X$ in the assessment of AD risk, where $Z$ was identified as a more significant than $X$ for 107 out of 130 SNPs in the ADNI dataset and 46 out of 63 SNPs in the BICWALZS dataset, representing 79.3% of the total. The results of the statistical analysis also presented that the combined effect of NetPRS demonstrated a 33.8% greater capacity to discriminate AD patients than the independent effect on average.

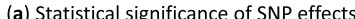

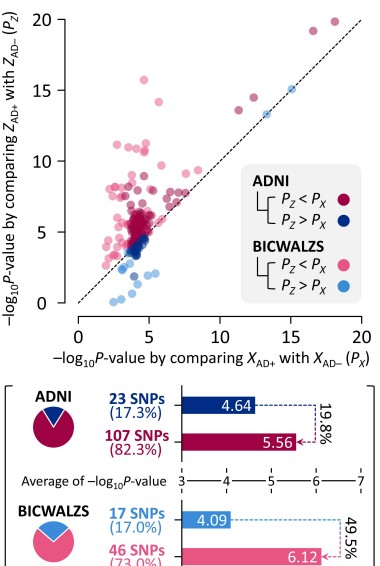

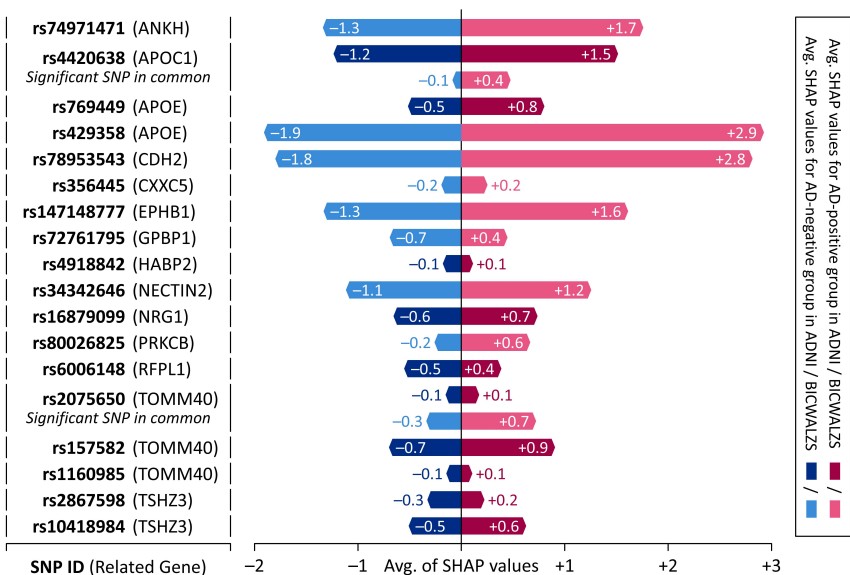

Fig. 4. Results for model interpretation on NetPRS. (a) compares the statistical significance of the independent and combined SNP effects for the AD-positive and AD-negative groups. (b) depicts the comparison of SHAP values for the top-10 most significant SNPs in the ADNI and BICWALZS.

Next, in the explanatory analysis, we identified the impact of SNPs in the combined effect by applying SHapley Additive exPlanations (SHAP) [36]. Fig. 4(b) shows the group-wise comparison of SHAP values for the top-10 most significant SNPs, denoted as key SNPs, in the ADNI and BICWALZS datasets. The SNP with the highest impact on AD risk was rs429358, which is related to APOE, followed by rs78953543, which is associated with CDH2 that has been identified to accelerate β-amyloid-triggered synapse damage [37]. Moreover, there were two common key SNPs, rs4420638 and rs2075650, that are associated with APOE-neighboring genes, APOC1 and TOMM40, respectively, which are significantly related to AD risk [38]. The key SNPs associated with ANKH, EPHB1, and NECTIN2 which are implicated in AD pathogenesis [39-41], also showed high impacts on predicting AD risk by NetPRS.

As a result, in AD risk assessment, the combined effect of SNPs by NetPRS was more effective than the independent effect of SNPs in distinguishing AD patients with greater clarity, with the impact of the identified key SNPs aligning with the previous clinical studies. Our findings suggest that the combined effect represents the collective influences of SNPs through their interactions, encompassing the comprehensive information on the phenotypic and genomic interactions.

## V. CONCLUSION

In this study, we propose a novel method, NetPRS, for AD risk assessment by utilizing the interactive effects between SNPs. The pronouncing features of NetPRS are summarized in three points. First, NetPRS quantifies the interactions that SNPs can exert through their related genes and GGIs. These genomic interactions provide comprehensive information on the SNP interactions along with the existing epistatic effects. Second, NetPRS derives the global interactive effect from the SNP networks. By propagating the independent effect through the networks, NetPRS extracts the effects arising from

infinite-order interactions. This approach significantly improves computational complexity compared to methods that use combinations of SNPs as variables. Third, NetPRS integrates various SNP effects to assess AD risk. In this process, the optimal integrating ratio of each effect is derived for the most accurate prediction. The proposed method provides an intuitive interpretation for output results by linearly combining each effect without non-linear transformation. Experimental results showed that learning both phenotypic and genomic interactive effects together with the independent effect yielded the most accurate AD risk assessment. Several analyses for the model interpretation confirmed that the advantage of NetPRS results from that the combined effect differentiates AD groups more clearly and reflects the genetic combinations significantly affecting AD pathogenesis.

Here are some remarks on the proposed method. First, it is possible that a more optimized construction of SNP interaction network may improve the utility of NetPRS. The computational burden associated with fitting logistic models to every pair of SNPs becomes significantly more demanding as the number of SNPs increases. Although this issue was mitigated by a pre-filtering process that employed statistical significance testing of SNPs through GWAS, future work will concentrate on the optimization of NetPRS through the use of graphical models, as well as the identification of the optimal $P$-value threshold for SNPs during the training process, beyond the current empirical search. Second, the integration of biological domain knowledge has the potential to enhance the validity of NetPRS. Further validation across multiple databases, along with expansion to other channels such as DNA and RNA in addition to genes, will facilitate a more sophisticated representation of genomic interactions between SNPs. Therefrom, a follow-up model for NetPRS is currently under development, which applies the biologically-informed clustering, with the expectation of providing further clarification regarding the impact of SNPs and those interactions on the AD risk.

TABLE A1. RESULTS FOR PERFORMANCE COMPARISON

| Method | ADNI Dataset | | | BICWALZS Dataset | | |
|--------|---------|---------|---------|---------|---------|---------|
| | Level 1 | Level 2 | Level 3 | Level 1 | Level 2 | Level 3 |
| wPRS | 0.5439 | 0.5766 | 0.6391 | 0.5018 | 0.5235 | 0.6040 |
| PRSice2 | 0.5993 | 0.6381 | 0.6833 | 0.5271 | 0.5395 | 0.6282 |
| EBPRS | 0.6036 | 0.6352 | 0.6946 | 0.5451 | 0.5546 | 0.6354 |
| ERS | 0.6441 | 0.7260 | 0.6152 | 0.6262 | 0.6830 | 0.6137 |
| PRSi | 0.6526 | 0.7364 | 0.6282 | 0.6432 | 0.6904 | 0.6280 |
| hiPRS | 0.6594 | 0.7372 | 0.6323 | 0.6418 | 0.7079 | 0.6337 |
| NetPRS | 0.7319 | 0.7561 | 0.7281 | 0.7084 | 0.7342 | 0.7045 |

TABLE A2. RESULTS FOR ABLATION STUDY

| Method | ADNI Dataset | | | BICWALZS Dataset | | |
|--------|---------|---------|---------|---------|---------|---------|
| | Level 1 | Level 2 | Level 3 | Level 1 | Level 2 | Level 3 |
| I | 0.5018 | 0.5235 | 0.6040 | 0.6928 | 0.7162 | 0.7070 |
| P | 0.5271 | 0.5395 | 0.6282 | 0.7194 | 0.7373 | 0.7158 |
| G | 0.5451 | 0.5546 | 0.6354 | 0.7212 | 0.7325 | 0.7085 |
| I + P | 0.6262 | 0.6830 | 0.6137 | 0.7293 | 0.7438 | 0.7184 |
| I + G | 0.6432 | 0.6904 | 0.6280 | 0.7301 | 0.7394 | 0.7139 |
| P + G | 0.6418 | 0.7079 | 0.6337 | 0.7299 | 0.7460 | 0.7204 |
| NetPRS | 0.7084 | 0.7342 | 0.7045 | 0.7319 | 0.7561 | 0.7281 |

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
