# NetPRS: SNP interaction aware network-based polygenic risk score for Alzheimer's disease

## SUPPLEMENTARY MATERIAL

### TABLE S1
#### RESULTS FOR PERFORMANCE COMPARISON ON ADNI DATASET

| Method | Level 1 | Level 2 | Level 3 |
|--------|---------|---------|---------|
| wPRS | 0.5439 | 0.5766 | 0.6391 |
| PRSice2 | 0.5993 | 0.6381 | 0.6833 |
| EBPRS | 0.6036 | 0.6352 | 0.6946 |
| ERS | 0.6441 | 0.7260 | 0.6152 |
| PRSi | 0.6526 | 0.7364 | 0.6282 |
| hiPRS | 0.6594 | 0.7372 | 0.6323 |
| NetPRS | 0.7319 | 0.7561 | 0.7281 |

### TABLE S2
#### RESULTS FOR PERFORMANCE COMPARISON ON BICWALZS DATASET

| Method | Level 1 | Level 2 | Level 3 |
|--------|---------|---------|---------|
| wPRS | 0.5018 | 0.5235 | 0.6040 |
| PRSice2 | 0.5271 | 0.5395 | 0.6282 |
| EBPRS | 0.5451 | 0.5546 | 0.6354 |
| ERS | 0.6262 | 0.6830 | 0.6137 |
| PRSi | 0.6432 | 0.6904 | 0.6280 |
| hiPRS | 0.6418 | 0.7079 | 0.6337 |
| NetPRS | 0.7084 | 0.7342 | 0.7045 |

### TABLE S3
#### RESULTS FOR ABLATION STUDY ON ADNI DATASET

| Model | Level 1 | Level 2 | Level 3 |
|-------|---------|---------|---------|
| I | 0.6928 | 0.7162 | 0.7070 |
| P | 0.7194 | 0.7373 | 0.7158 |
| G | 0.7212 | 0.7325 | 0.7085 |
| I + P | 0.7293 | 0.7438 | 0.7184 |
| I + G | 0.7301 | 0.7394 | 0.7139 |
| P + G | 0.7299 | 0.7460 | 0.7204 |
| NetPRS | 0.7319 | 0.7561 | 0.7281 |

### TABLE S4
#### RESULTS FOR ABLATION STUDY ON BICWALZS DATASET

| Method | Level 1 | Level 2 | Level 3 |
|--------|---------|---------|---------|
| I | 0.6706 | 0.6955 | 0.6841 |
| P | 0.6962 | 0.7160 | 0.6925 |
| G | 0.6980 | 0.7113 | 0.6855 |
| I + P | 0.7058 | 0.7223 | 0.6951 |
| I + G | 0.7066 | 0.7180 | 0.6908 |
| P + G | 0.7064 | 0.7244 | 0.6970 |
| NetPRS | 0.7084 | 0.7342 | 0.7045 |