# OpenReview forum: "NetPRS: SNP interaction aware network-based polygenic risk score for Alzheimer’s disease"
_IEEE.org/EMBS/BHI/2024/Conference — IEEE BHI'24_

### Official Review · Reviewer_CKpK · 2024-08-01
**NetPRS: SNP interaction aware network-based polygenic risk score for Alzheimer’s disease**

**Overall Rating:** 8
**Confidence:** 3

**Other Quality Metrics:**

Claritiy of Writing: Great
Clinical Significance: Excellent
Methodological Novelty: Great
Experiments and Results: Excellent

**Questions For The Authors:**

I do not have any questions. If there is a public data set than the one used in the study adding its results like Figure 4a will improve the credibility of the model.

**Strengths:**

The novelty of the method is explained well and its comprehensive experimental study section supports the conclusion. The paper is clearly written although it has many mathematical equations. Although I am not a genomic expert I understood it well.

**Summary Of The Paper:**

The study develops a machine learning model using the independent and interactive effects of SNPs to generate a risk score for Alzheimer disease.

**Weaknesses:**

Some abbreviations are left unexplained such as SHAP and gene names and snp names makes it difficult to read the results section. Please explain them for a general audience.

---

### Official Review · Reviewer_2ZcB · 2024-08-14
**NetPRS: SNP interaction aware network-based polygenic risk score for Alzheimer’s disease**

**Overall Rating:** 6
**Confidence:** 4

**Other Quality Metrics:**

(a) Clarity of Writing: Good
(b) Clinical Significance: Excellent
(c) Methodological Novelty: Great
(d) Experiments and Results: good

**Questions For The Authors:**

1. How do the authors anticipate NetPRS performing in populations with different genetic profiles? Have the authors considered applying the model to diverse cohorts, or are there specific reasons for focusing solely on the Korean cohort?
2. In the explanatory analysis, the authors conclude that SNPs with a high impact on NetPRS exert their influence collectively through interactions, yet the discussion primarily focuses on the roles of these key SNPs at an individual level. Are there any results from this study or previous findings that support the collective impact of these SNPs? Could the authors provide further clarification or evidence for this statement?

**Strengths:**

The manuscript provides a detailed and rigorous approach to constructing SNP-interaction networks and effectively using them for Alzheimer’s disease risk prediction. The thorough experimental evaluation demonstrates significant improvements in prediction accuracy with NetPRS compared to traditional PRS models. Additionally, the inclusion of an ablation study is commendable, as it highlights the contributions of the various components of the model, further validating the method’s robustness and effectiveness.

**Summary Of The Paper:**

The manuscript presents NetPRS, a novel polygenic risk score model for Alzheimer’s Disease (AD) that incorporates both phenotypic and genomic interactions between SNPs. Unlike traditional models that focus solely on independent SNP effects, NetPRS uses a two-stage approach to construct SNP interaction networks and predict AD risk. When evaluated on a Korean cohort, NetPRS demonstrates significant improvements in prediction accuracy compared to a conventional weighted-PRS model, attributed to its ability to capture complex SNP interactions. The manuscript also includes an explanatory analysis that highlights the contributions of individual SNPs, enhancing the interpretability of the model.

**Weaknesses:**

While the results on the Korean cohort are promising, the paper does not sufficiently address the potential limitations in generalizability. Given the significant variation in genetic architecture across populations, it is crucial to explore how the model might perform in diverse cohorts, particularly since Alzheimer’s disease risk factors can differ globally. If obtaining SNP data from other populations is challenging, the authors are recommended to conduct resampling tests on their validation cohort or use K-fold cross-validation to demonstrate the robustness of their model. Additionally, although the authors acknowledge the challenges of computational complexity, particularly with higher-order SNP interactions, the paper lacks a thorough comparison and detailed discussion of this issue. Furthermore, while the paper compares NetPRS with a conventional weighted PRS model, it does not provide a comparison with other state-of-the-art approaches that also consider SNP interactions, such as the method referenced in their introduction (Reference 17, DOI: https://doi.org/10.1186/s13195-021-00794-8). Including such comparisons would strengthen the argument for NetPRS and provide a more comprehensive evaluation of its effectiveness.

---

### Official Review · Reviewer_7mX5 · 2024-08-16
**The novel method incorporates SNP interaction networks into polygenic risk scoring.**

**Overall Rating:** 7
**Confidence:** 4

**Other Quality Metrics:**

Clarity of writing: great
Clinical Significance: fair
Methodological Novelty: great
Experiments and Results: great

**Questions For The Authors:**

1. For the two SNP interaction networks, Phenotypic Interaction Network and Genomic Interaction Network, how do you interpret the values for the edges? Are there any meaningful statistics (e.g., marginal/conditional correlation) for the edge in the network structure?
2. When constructing the Phenotypic Interaction Network, fitting logistic regression for each pair of SNPs individually can become computationally intensive as the number of SNPs increases. Is there any way to deal with this problem?
Also, alternative methods, such as Gaussian graphical models or graphical lasso, may offer more efficient approaches for network construction.
3. Why do you need to use data from dbSNP and STRING for the real data analysis? What specific information do these two databases provide in your experiment?
4. For the comparative analysis, although Level 3 shows the best performance, it has only two edges in both networks. Does a network with just two edges truly capture significant information?
5. Based on the experiment, we observed that SNP thresholding significantly impacts model performance. How can we determine the appropriate number of SNPs for the analysis?
6. When calculating the PRS, it is important to account for Linkage Disequilibrium. Have you considered this in your model, or is there any part already addressing it?

**Strengths:**

1. The study considers phenotypic and genomic SNP interactions, addressing the limitations of traditional PRS models that consider only independent SNP effects.
2. NetPRS is found to be 35% better than the conventional PRS models for predicting the risk of Alzheimer’s disease.
3. The ablation study conducts a comprehensive analysis to investigate the contributions of different SNP effects (independent, phenotypic, and genomic) to the model’s performance, providing an understanding of their impact on AD risk prediction.

**Summary Of The Paper:**

The study presented a novel approach named Network-based Polygenic Risk Score (NetPRS) for estimating Alzheimer's disease risk. Most of the existing PRS models are based on the main effects of SNPs, while NetPRS considers both phenotypic and genomic SNP interactions. The method first builds SNP interaction networks and then extracts independent effects across these networks by a graph-based machine-learning model. This allows NetPRS to obtain global interactive effects between SNPs, which is then added to independent effects to estimate AD risk. The findings demonstrate that NetPRS achieves superior accuracy in risk estimation compared to standard PRS models.

**Weaknesses:**

1. The model is relatively complex, and the computational cost of constructing and propagating the network makes it difficult to extend the analysis to larger datasets.
2. The values for the smoothness parameter, p-value thresholds, and the number of SNPs are determined too subjectively. A more scientific approach to deciding these values is required.
3. The paper only focuses on Korean data but does not extensively explore the performance of NetPRS on external datasets or other ethnic groups.

---

### Decision · Program_Chairs · 2024-09-23

Accept